# CAD/CAM Resin-Based Composites for Use in Long-Term Temporary Fixed Dental Prostheses

**DOI:** 10.3390/polym13203469

**Published:** 2021-10-09

**Authors:** Franziska Hensel, Andreas Koenig, Hans-Martin Doerfler, Florian Fuchs, Martin Rosentritt, Sebastian Hahnel

**Affiliations:** 1Department of Dental Prosthetics and Materials Science, Leipzig University, 04103 Leipzig, Germany; Franziska.Hensel@medizin.uni-leipzig.de (F.H.); hans-martin.doerfler@htwk-leipzig.de (H.-M.D.); Florian.Fuchs@medizin.uni-leipzig.de (F.F.); sebastian.hahnel@medizin.uni-leipzig.de (S.H.); 2Department of Mechanical and Energy Engineering, University of Applied Sciences, 04277 Leipzig, Germany; 3Department of Prosthetic Dentistry, Regensburg University Medical Centre, 93053 Regensburg, Germany; martin.rosentritt@klinik.uni-regensburg.de

**Keywords:** three-unit FDP, surface wear, dimethacrylats, chewing simulation, RBC, micro X-ray computer tomograph, confocal laser scanning microscope

## Abstract

The aim of this in vitro study was to analyse the performance of CAD/CAM resin-based composites for the fabrication of long-term temporary fixed dental prostheses (FDP) and to compare it to other commercially available alternative materials regarding its long-term stability. Four CAD/CAM materials [Structur CAD (SC), VITA CAD-Temp (CT), Grandio disc (GD), and Lava Esthetic (LE)] and two direct RBCs [(Structur 3 (S3) and LuxaCrown (LC)] were used to fabricate three-unit FDPs. 10/20 FDPs were subjected to thermal cycling and mechanical loading by chewing simulation and 10/20 FDPs were stored in distilled water. Two FDPs of each material were forwarded to additional image diagnostics prior and after chewing simulation. Fracture loads were measured and data were statistically analysed. SC is suitable for use as a long-term temporary (two years) three-unit FDP. In comparison to CT, SC featured significantly higher breaking forces (SC > 800 N; CT < 600 N) and the surface wear of the antagonists was (significantly) lower and the abrasion of the FDP was similar. The high breaking forces (1100–1327 N) of GD and the small difference compared to LE regarding flexural strength showed that the material might be used for the fabrication of three-unit FDPs. With the exception of S3, all analysed direct or indirect materials are suitable for the fabrication of temporary FDPs.

## 1. Introduction

Temporary fixed dental prostheses (FDPs) are essential for the success of prosthetic treatments. They are used to protect the prepared teeth from chemical, thermal, and bacterial irritations, restore form and function, and may be used to visualize the design of the planned definitive restoration. In addition, they may also help to shape marginal gingival areas [1]. Temporary FDPs should be biocompatible and easy to process, feature a high accuracy of fit and sufficient stability, have low manufacturing costs and a suitable aesthetic appearance [2,3,4].

Depending on the indication and the intended time in clinical service, a variety of materials and manufacturing techniques are available. According to the fabrication technique, temporary FDPs can be divided into direct and indirect restorations. The various technologies have an influence on the individual wearing time. Temporary FDPs fabricated using the direct technique are recommended for a wearing time between one and three months, while temporary FDPs fabricated using indirect techniques can be in service for up to two years [5,6]. Particularly in clinical settings requiring alterations in the vertical or horizontal dimension of occlusion, extended simulation with long-term temporary restorations are mandatory and also required for forensic reasons.

Most temporary FDPs are fabricated in a chairside process using an overimpression technique in combination with autopolymerising resin-based materials. These FDPs are associated with various shortcomings that result from unfavourable conditions during the manufacturing process. Inhomogeneities, like voids or contaminations from the oral cavity, may lead to discoloration, poor mechanical properties, and insufficient fit [7]. Especially for temporary FDPs, mechanical interactions and the fluctuation of temperature in the oral cavity cause stress, which may result in a failure of the interim restoration. Repair or fabrication of a new interim restoration require additional time and increase treatment costs [8].

Computer-aided design and computer-aided manufacturing (CAD/CAM) technologies in combination with industrially manufactured polymer blocs or discs may solve some of these issues. Due to optimized and standardized industrial polymerization conditions at higher temperatures (>50 °C) and a longer duration, the degree of polymerisation is increased in these materials in comparison to direct resin-based materials. The high pressure allows the inclusion of high filler contents and therefore to generate a microstructure with few imperfections and homogenous properties. As a result, resin-based CAD/CAM materials feature improved mechanical properties and biocompatibility as well as less biofilm formation on their surface than direct resin-based materials. [9,10,11,12,13]. While these materials are processed indirectly, the digital workflow offers various advantages such as a fast-manufacturing process and the opportunity to duplicate the restoration in case of loss or failure. Moreover, modifications of the temporary FDPs in therapy sequences are simple to perform.

Resin-based CAD/CAM materials are commonly divided into three groups:CAD/CAM polymers on the basis of polymethyl methacrylate (PMMA) resins with low inorganic filler contents [14];highly filled CAD/CAM resin-based composites (RBCs) based on dimethacrylates (DMA) [15], andresin-filled hybrid ceramics based on “polymer infiltrated ceramic network” (PICN) [16].

CAD/CAM polymers are based on methyl methacrylates (MMA) and mostly a low content (up to 10 wt%) of inorganic fillers [14]. With a flexural strength between 80–160 MPa and a modulus of elasticity between 2 and 5 MPa [17], these polymers are primarily suitable for the fabrication of temporary FDPs.

CAD/CAM RBCs are based on different DMA such as Bisphenol A-Glycidylmethacrylate (Bis-GMA), Urethanedimethacrylate (UDMA), and Bisphenol-Dimethacrylate (Bis-DMA) and feature a high content (61–86 wt%) of inorganic fillers [15]. With a flexural strength between 130–200 MPa and a modulus of elasticity between 8–20 MPa [17], these materials have mechanical properties close to natural dentin (Table 1). CAD/CAM RBCs are primarily used for the fabrication of long-term temporary restorations. As a result of their composition, CAD/CAM RBCs produce less abrasion of enamel antagonists than ceramic materials [18,19]. The low modulus of elasticity and their composition allow resilient CAD/CAM materials to compensate destructive fracture energy by elastic and plastic deformation to a greater extent than stiffer ceramic CAD/CAM materials [20]. The plastic deformability produces a depressant, comfortable, and natural chewing feeling for the patient [20,21]. In the case of PMMA, these properties coincide with higher material wear than in highly filled CAD/CAM RBCs or resin-filled hybrid ceramics [22].

Temporary FDPs fabricated from PMMA feature favourable aesthetic properties because of their refractive index and can be easily customised. They also show a lower tendency towards discolouration than CAD-CAM RBCs [23]. It has been reported that resin-based materials tend to absorb liquids and discolour in the long term [24]. With regard to this aspect, Bis-GMA features lower colour stability than UDMA as a result of its increased water absorption and solubility properties [25]. Stawarczyk et al. showed that after a storage period of 180 days in various colouring liquids most CAD/CAM RBCs had s similar colour stability as ceramics [23]. Nevertheless, potential discolouration is a limiting factor for the application of resin-based materials as definitive restoration. Both materials are suitable to be applied as long-term temporary FDPs, even in extended restorations. Especially in cases with extended restorations, their low density and low weight improve the wearing comfort for the patients. Due to their low modulus of elasticity and brittleness, polymer-based CAD/CAM materials also have favourable properties for the treatment of patients with bruxism, even if it is not commonly included in the indications [14]. While both materials are approved for the fabrication of temporary FDPs, their properties might also allow application in definitive restorations. However, clinical investigations addressing the long-term stability of tooth-coloured polymer-based CAD/CAM materials for multi-unit FDPs are currently rare. Against this background, the current study analysed a newly developed CAD/CAM RBC material regarding its long-term stability in multi-unit FDPs in comparison to several other currently available materials. Another CAD/CAM RBC authorized for the fabrication of single crown FDPs only was also analysed as its mechanical properties indicate a potential application in multi-unit FDPs. The null hypothesis was two-fold: The mechanical stability of temporary 3-unit FDPs is independent of the manufacturing process (1), and all materials of each respective groups (i.e., for application in either temporary or definitive restorations) feature an identical behaviour (2).

## 2. Materials, Experimental Procedure and Methods

### 2.1. Materials

In the current study, three CAD/CAM resin-based composites (SC, CT, GD) were compared with two direct resin-based materials (LC, S3). Both groups can be used for temporary purposes. In addition, a CAD/CAM ceramic (LE) for permanent application was used for reference purposes (Table 2).

### 2.2. Experimental Procedure

For each material (LC, S3, SC, CT, LE, GD), 20 three-unit fixed dental prostheses (FDP) were manufactured. Therefore, two resin teeth (24, 26; Kavo Dental, Biberach, Germany) were prepared for supply with a FDP. In a second step, roots made of wax were added to the bottom of the prepared resin teeth to simulate the anatomical shape of natural teeth. The teeth were then digitalized using a 3D scanner (inEOS X5; Software: inLab CAM Software; Dentsply Sirona Deutschland GmbH, Hessen, Germany) and the dataset was used as a template to mill (inLab MC X5; Dentsply Sirona Deutschland GmbH) wax teeth (Zirlux Wax, Henry Schein Dental Deutschland GmbH, Langen, Germany), which were then cast from a Co-Cr-Mo-alloy (remanium star, Dentaurum, Ispringen, Germany). The roots of the metal teeth were dipped in wax bath and positioned in resin blocks (Technovit 4000, Kulzer GmbH, Hanau, Germany). In a second fabrication process, the wax was replaced by a 1 mm polyether layer (Impregum, 3M Deutschland GmbH, Seefeld, Germany) to simulate the resilience of the human periodontium.

For indirectly processed materials, the prepared teeth were digitalized (inEOS X5, inLab CAM Software) and three-unit FDPs with identical outer dimensions were designed (inLab CAM Software) with a minimum thickness of 1.0 mm circular and 1.5 mm occlusal. For directly processed materials, FDPs were produced using a silicone form (HS-A Silikon putty soft and light body, VPS Hydro, Henry Schein Dental Deutschland GmbH, Langen, Germany) that was moulded on a tooth model supplied with one of the milled FDPs.

The metal teeth and the inner surface of the FDPs were sandblasted (teeth: Al_2_O_3_, 150 µm, 4.0 bars; restorations: Al_2_O_3_, 50 µm, 1.5–2.0 bar, exception: Structur 3). The three-unit FDPs were treated with a bonding agent (Monobond Plus, Ivoclar Vivadent AG, Schaan, Liechtenstein) and adhesively bonded to the teeth (Bifix QM, VOCO GmbH, Cuxhaven, Germany) in accordance with the manufacturers’ instructions (Table 3).

As the combined use of chewing simulation and thermocycling is a well-documented method to mimic clinical situations [30], 10/20 FDPs were exposed to thermal and mechanical loading. Based on the assumption that a chewing simulation with 1.2 million cycles corresponds to a wearing time of approximately five years [31], a clinical use of two years for a long-term temporary restoration was simulated in this study. The use of thermocycling made it possible to simulate intraoral temperature fluctuations and to expose all test specimens to a standardised and reproducible load [32,33].

For the attachment of all FDPs, the same permanent cement was used. Apart from avoiding premature loosening during chewing simulation which might have occurred when using a temporary cement, this procedure helped to exclude other potentially influencing factors regarding a reduction or an increase of fracture strength that might been have been caused by the application of different cements.

For better comparability, the same parameters were set for all FDPs in terms of wall, occlusal, and connector thickness. The largest value identified for the minimum requirements issued by the manufacturers was defined as guideline value. In order to perform artificial aging, all specimens were stored in distilled water for 24 h prior to mechanical testing.

The study was divided into an experimental and a control group. Two FDPs from each material in the experimental group were randomly selected for additional image diagnostics prior and after chewing simulation (Figure 1).

### 2.3. Methods

All restorations in the experimental group (n = 60) were acceleratedly stressed by chewing simulation (CS-4.8, SD Mechatronik GmbH, Feldkirchen-Westerham, Germany; 480,000 cycles, 50 N, 1.3 Hz) with integrated thermal cycling (TC: 1200 cycles between 5 °C and 55 °C, 2 min for each cycle, H_2_O = demineralised). Enstatite balls (Ø 6 mm, CeramTec GmbH, Plochingen, Germany) served as antagonists and were positioned in occlusal contact to the pontic of the three-unit FDPs. Failures were documented and failed samples were excluded from the further process.

With the exception of the samples used for the imaging procedure, all other samples as well as control group were loaded to fracture in the universal testing machine (ZwickRoell Retroline, ZwickRoell, Ulm, Germany). The universal testing machine was combined with HBM measuring amplifier systems (MGCplus, HBM, Darmstadt, Germany) and high-resolution inductive displacement transducers (WI/2mm-T, HBM, Darmstadt, Germany), which allow quantification of the deflection in the middle of the FDP depending on the force applied. The force was applied in the centre of the pontic using a steel sphere (Ø 6 mm, cross-head speed 1 mm/min). A 0.5 mm thick tin foil (Renfert, Hilzingen, Germany) was inserted between restoration and sphere to prevent force peaks. The maximum fracture load and the deformation were measured until the material failed. The failure detection was set to 50% loss of the maximum loading force. The failure mode of all restorations was documented. The specimens were optically examined after fracture testing for a third time.

Calculations of mean values and standard deviation (SD) and statistical analyses were performed using SPSS Statistics analysis software (SPSS, IBM, v.25). Normality of data distribution was analysed using the Shapiro-Wilk test. For the fracture load values, the homogeneity of variances was first checked using Levene’s test and then analysed using the T-test for independent samples or Mann-Whitney U-test. Kruskal-Wallis test and Bonferroni-Dunn post-hoc test were used to evaluate the different surface roughness values. Significance level was set at α = 0.05. Failed or debonded FDPs were excluded from statistical evaluation.

A 3D laser scanner (inEOS X5; software: inLab CAM and GOM inspect sofware) was used to determine the non-reversible deformation and a confocal laser scanning microscope with 10× magnification (VK-X1000/1050, Keyence, Neu-Isenburg, Germany; Nikon CF IC EPI Plan 10×) was applied to image the surface and to quantify the surface roughness. In order to take a potential directional texture of the mechanical load into account, 20 profile lines with an interval of 20 px in each direction and a length of 1 mm were measured with an orthogonal arrangement. After applying a profile filter with a cut-off wavelength of λs 2.5 µm and λc 0.25 mm (end effect correction, filter type: double Gaussian), the arithmetical mean height (Ra) and the maximum height (Rz) of the roughness profiles were determined. Three areas of the occlusion surface were analysed, including the unstressed surface (0) prior to and the stressed surface (X) after chewing simulation. The stressed surface was divided into an upper surface (X_p: contact/pressure point of the antagonist with the specimen) and a lower surface (X_w: downward movement of the antagonist into the central fossa).

In order to identify changes within the structure of the FDPs (like micro cracks, air voids), the samples were investigated with an industrial micro X-ray computer tomograph (µXCT, prototype, FhG Dresden, Germany). The directional X-ray tube (FXE 225.99 YXLON International GmbH) was applied with X-ray power of 24 Watt (beam energy 180 kV and flux 160 μA) and a copper filter of 0.1 mm Cu. A special sample holder made of carbon (in the beam path) and aluminum (out the beam path) was developed, which allowed analysis of the connector area between the two tooth stumps prior and after chewing simulation in identical samples. With the X-ray tube and the 2D-detector (1621xN, PerkinElmer Inc., Waltham, MA, USA) and a step size of 0.45/360 (800 positions), a resolution of 7.9 μm (V = 493 μm³) was achieved. The data processing and analysis are explained in detail in [34].

## 3. Results

### 3.1. Fracture Loads and Failure Rates

Only FDPs fabricated from S3 fractured during the time lapse loading by simultaneous chewing simulation and thermocycling after ≤130,718 cycles. The failure rate was 50 % (i.e., 5/10 FDPs). The mean values of the failure loads for the FDPs fabricated from the various materials are graphically displayed with standard deviations in Figure 2. The FDPs fabricated from SC had fracture loads similar to those of S3 and LC. Lowest values were identified for FDPs fabricated from CT. With the exception of LC, higher failure loads were measured after chewing simulation and thermocycling in comparison to the control group.

For FDPs fabricated from CT and GD, fracture loads were significantly higher after chewing simulation and thermocycling (*p* < 0.001 and *p* < 0.05, respectively) (Table 4). The FDPs fabricated from the indirectly processed materials SC and CT, which have the same indication, showed significant differences in failure loads (*p* < 0.001), with SC having a significantly higher fracture load than CT.

After chewing simulation and thermocycling, significant differences were identified between the temporary indirectly processed materials CT and SC (*p* < 0.05) and the permanent indirectly processed materials GD and LE (*p* < 0.05).

### 3.2. Further Studies to Clarify the Mechanism of Action

#### 3.2.1. Surface Wear

Mechanical loading caused relevant wear on the surface of the FDPs, yet differences in wear rates were identified between the various materials (Figure 3).

Highest vertical substance loss was 1.52 mm, which was identified for S3; lowest wear was 0.05/0.07 mm for LE. For the FDPs fabricated from SC, a similar amount of wear of 0.55/0.81 mm could be detected as for similar polymer-based products for the fabrication of indirect restorations (CT, GD) (Table 5).

There were also clear differences in the geometry of the wear facets in the enstatite antagonists (Figure 4). Greatest changes in geometry were identified for LE and least changes for SC and S3.

#### 3.2.2. Surface Analyses

Statistical analyses showed that almost all Ra (arithmetical mean roughness value) and Rz (maximum roughness heigth) values increased significantly (*p* < 0.05 or *p* < 0.001) (Table 6). There was no significant change in the Ra value for the material LE between the surfaces X_p and X_w. There were no significant changes in Ra and Rz between the X_p and X_w surfaces for S3 and GD. Also, no significant changes in Rz were identified between 0 and X_p for SC and in Ra and Rz between X_p and 0 for LC.

#### 3.2.3. Microstructure

The cross-sectional images of the respective pontic areas showed relevant differences in their microstructure. Radiopaque fillers (<100 µm) in SC, larger air voids (<600 µm) in LC, and presumably radiopaque (light grey values) residues of the burs on the surface of CT could be identified (Figure 5). The light semicircle in the lower centre in LE is a ring/hardening artefact and not a local change in the microstructure caused by the chemical composition in the material (large atomic mass). This assumption is confirmed by the light surface edge. Chewing simulation and thermocycling did not cause any changes such as microcracks in the microstructure.

## 4. Discussion

The hypothesis that the mechanical strength of temporary three-unit FDPs is independent of the manufacturing process could only partially rejected and the hypothesis that all investigated materials of the respective group (temporary and permanent) show an identical behaviour could be rejected.

### 4.1. Mechanical Behavior

Fracture loads showed statistically significant differences between but also within the different indication groups. For directly-processed temporary materials, no significant differences were identified between the materials. With regard to fracture load, statistically significant differences were identified between GD and LE. These results might be due to the different composition of the materials and the resulting material properties (Table 1). As expected, zirconia (LE) showed the highest fracture load values (0: 1542 ± 646 N/X: 1706 ± 248 N), followed by CAD/CAM resins-based composites (GD) (0: 1099 ± 150 N/X: 1327 ± 177 N), which is currently only approved for the fabrication of single tooth permanent FDPs. The small differences in fracture load in contrast to the significant differences in flexural strength (LE 1200 vs. GD 155 MPa, Table 1) as well as the absence of failures, however, indicate that the flexural strength cannot be used as the sole parameter for defining the clinical indication of a material. Thus, GD might be employed for the fabrication of definitive three-unit FDPs, too. However, further tests with a higher number of chewing cycles are necessary to corroborate this assumption prior to performing clinical studies.

In the group of indirectly processed temporary CAD/CAM materials, statistically significant differences in fracture loads were identified between CT and SC. Unlike all other resin-based materials, CT is based on polymethyl methacrylate (PMMA) rather than DMA and is micro-filled. FDPs fabricated from CT showed the lowest breaking load values in this study in comparison to the other materials. These results might be explained by the fact that PMMA is a thermoplastic and, in contrast to thermosets (DMA), has significantly fewer cross-links, which results in poorer mechanical behaviour [17]. In addition, CT features a lower filler content (14.0 wt%) than SC (28.8 wt%) based on own thermogravimetric measurement), which might also explain its worse mechanical performance.

CT has the lowest flexural strength compared to the other materials (Table 1), which responds to the lowest breaking loads identified in the current study. Its modulus of elasticity is also lower than in the other materials (Table 1), which would result in higher deformation under cyclical mechanical loading during chewing simulation and might serve as an explanation why no failures were observed in FDPs fabricated from CT despite its low strength.

The directly processed temporary materials S3 (0: 805 ± 95 N/X: 850 ± 345 N) and LC (0: 943 ± 217 N/X: 756 ± 392 N) had similar fracture loads as the CAD/CAM material SC (0: 823 ± 148 N/X: 875 ± 190 N), yet 50 % of the FDPs fabricated from S3 failed during laboratory aging. The FDPs failed after ≤130,718 cycles in the chewing simulator, which corresponds to a time in clinical service of approximately six months. According to the manufacturer, the material is approved for application in long-term temporary restorations. The manufacturer of S3 defines the maximum wearing time as six months, whereas the manufacturer of LC issues a maximum wearing time of five years. However, S3 was the only material that partially failed during mechanical and thermal loading. This phenomenon may be due to limitations associated with the manufacturing process such as the inclusion of air voids or increased water absorption. However, µXCT measurements showed no microcracks or big air voids within the FDP (Figure 5, total porosity 0.25 vol%). The water absorption can negatively affect the durability of a resin-based dental restoration as it influences its dimensional stability and mechanical properties and acts like a plasticizer [33,35].

Apart from mechanical properties, there are different definitions regarding the required durability of long-term temporary restorations. Frequently, temporary dental restorations are used for weeks up to six months. In certain cases, e.g., in case of alterations in the vertical dimension of occlusion or occlusal adjustments, an extended period of up to two years may be necessary. The results of the current study underline that the requirements associated with the fixed temporary restoration should be carefully considered prior to treatment in order to choose the individually appropriate material. Apart from the high failure rate, S3 showed fracture load values similar to SC or LC.

Clinically observed occlusal forces, which usually occur during chewing processes, range from 12 to 90 N. However, occasionally bite forces may even be much higher. Previous studies revealed a mean maximum force in the molar region of 597 N in young healthy women and 847 N in men [36]. In the present study, the breaking forces were higher (>1000 N) in the group of the permanent materials. Nevertheless, fracture forces of S3, LC, and SC were close to the reported upper limit of reported maximum forces. Only CT showed statistically significantly lower values and therefore, the application of this material in long-term restorations can only be recommended with restrictions, e.g., in the anterior region.

Slightly higher fracture load values were identified after aging simulation with the exception of LC. This phenomenon might be due to the fact that the control group was stored in distilled water for 24 h prior to testing while the experimental group was stored in distilled water throughout the chewing simulation and thermocycling. A post-polymerisation process could be an explanation for the directly processed materials because of its higher monomer content; this thesis does, however, not explain similar observations in other materials, especially zirconia, which also showed higher forces after aging simulation.

It should be taken into consideration that mechanical stresses in the oral cavity differ between various individuals and depend on numerous variables such as tooth shape, occlusal contacts, or antagonist material, which cannot be extensively considered in a laboratory study.

### 4.2. Surface Properties

A favourable clinical performance of a provisional material is not exclusively dependent on its mechanical properties, but also on its interactions with the surrounding tissues. Therefore, factors such as marginal adaptation or colour stability should be analysed in future investigations. With regard to this aspect, surface wear is an important issue in the estimation of a dental material. In temporary restorations that are in clinical service for extended periods, wear might be a relevant problem as excessive abrasion of the material affects the occlusion and impairs function and stability of the restoration. This is especially true in settings where quadrants or entire jaws are restored and long-term temporary restorations are used to simulate the outcome and test adaptive coping.

With regard to material wear, highest abrasion was identified for S3 (1.52 mm). Similar wear rates in a clinical setting would have relevant consequences as they deteriorate the stability of the restoration and impair the vertical dimension of occlusion. LE showed the lowest wear, and only a gloss point was visible on the surface. The materials SC, CT, and GD showed similar wear values ranging between 0.55 mm and 0.87 mm. Tendentially, with increasing hardness of the material (Table 1), the abrasion of the antagonist increased (Figure 4) and maximum substance loss of the FDPs decreased (Table 5). The wear rates measured for LC, a directly processed temporary material, were surprising. With an abrasion of 0.42/0.56 mm it produced the third highest wear in the enstatite antagonist.

Surface analysis prior and after chewing simulation showed that, for most FDPs, there were significant changes or increases in roughness after chewing simulation. Highest roughness was identified for CT featured after mechanical loading. This phenomenon can be explained by its high polymer and low inorganic (filler) content, which based on Voigt’s model [37]—coincides with low hardness. In addition to that, the few cross-links in the polymer (PMMA) might also impair the wear resistance of the material.

With regard to material and antagonist wear, it should be mentioned that SC produced the lowest and LE and GD the highest wear in the enstatite antagonists. The reasons for the high antagonist wear include the higher hardness of the temporary materials in comparison to enstatite (Vickers hardness 530 based on Mohs hardness between 5.5).

### 4.3. Microstructure

As expected, the directly processed RBCs (S3 and LC) showed a higher grade of porosity than the indirectly processed materials (Figure 5). Micro-computed tomography images identified the biggest pores in LC, which probably respond to air pockets produced during the manufacturing process. SC, GD, and LE did not show any major defects (pores, blowholes, or cracks). The only notable feature were the larger radiopaque components in SC, which were not as clearly visible in any other material. The radiopaque bur debris on the surface, similar to the residues of dental burs used by the dentist for the preparation of teeth, may represent a problem with the biocompatibility due to their composition (e.g., tungsten carbide-cobalt in carbide burs) [38].

## 5. Conclusions

The current in vitro study includes the limitations of constant settings regarding wall, occlusal, and connector thickness. Moreover, only a limited number of potentially influencing parameters (e.g., distilled water, enstatite antagonist) can be simulated in a laboratory setting approaching the clinical reality. In this context, it must be borne in mind that the use of temporary cements in a chewing simulation setting might increase the risk of decementations, which is why permanent luting cements were used in the current trial. Based on these limitations, the null hypotheses could not be confirmed.

However, the following conclusions could be drawn:

Structur CAD (SC) is suitable for use in long-term temporary (two years) three-unit FDPs. In comparison to the indirectly processed material VITA CAD-Temp (CT), which is also used as temporary material, the breaking forces were significantly higher (SC > 800 N; CT < 600 N), the surface wear of the antagonists was lower, and wear of the FDP was similar.

Only DMA-based CAD/CAM RBCs with a high filler content should be used for the fabrication of long-term temporary FDPs that are in clinical service for more than six months.

The high breaking forces (1100–1327 N) of Grandio disc (GD) compared to the mean maximum chewing force in the molar region of 597 N in young healthy women and 847 N in men and the small difference compared to Lava Esthetic (LE) in relation to the flexural strength show that the material might be used for application in three-unit FDPs.

## Figures and Tables

**Figure 1 polymers-13-03469-f001:**
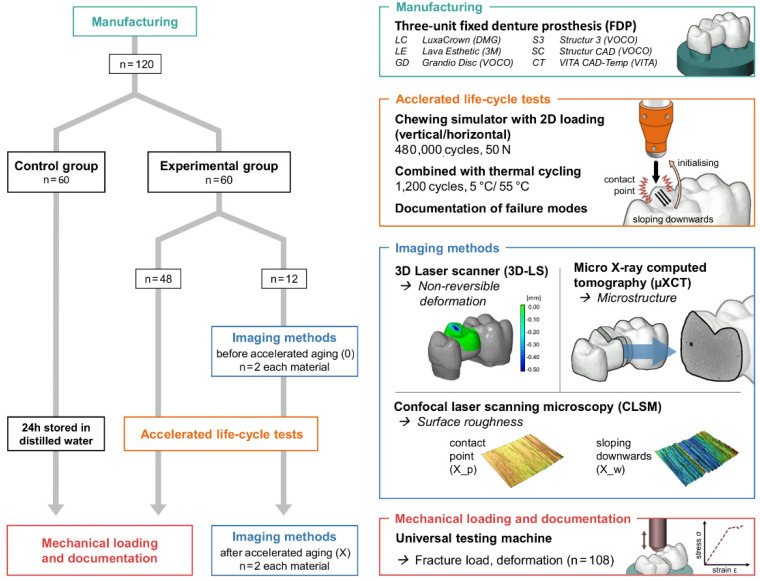
Flowchart of the test program with imaging techniques [micro X-ray computer tomograph (µXCT), confocal laser scanning microscope (CLSM, 2D), 3D scanner (3D-LS, 3D)] and universal testing machine.

**Figure 2 polymers-13-03469-f002:**
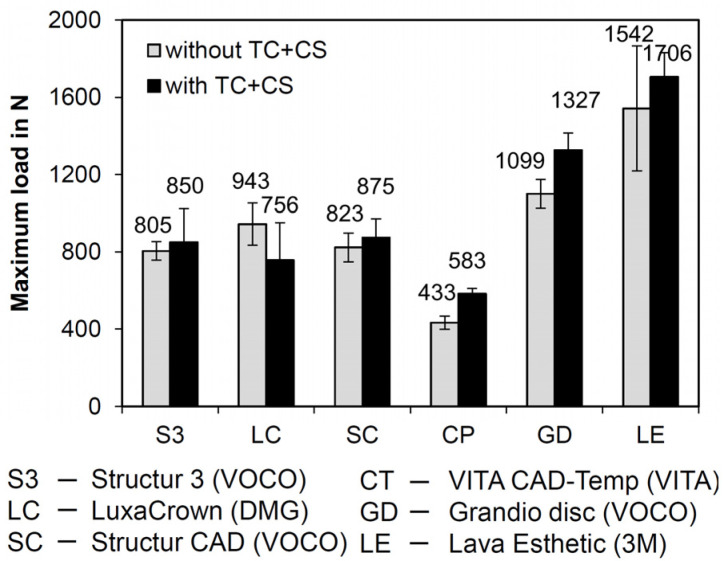
Failure loads without (10 samples per material) and after simultaneous chewing simulation and thermocycling (CS + TC) (8 samples per material; exception: S3; only four FDPs were forwarded to fracture analysis due to a 50% failure rate in chewing simulation).

**Figure 3 polymers-13-03469-f003:**
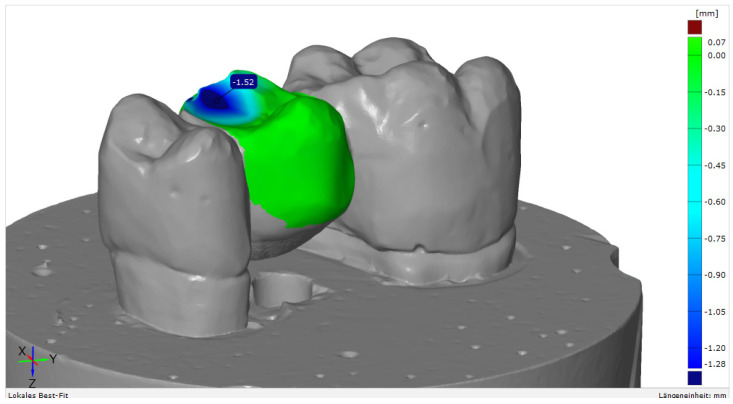
Example of a surface match analysis with GOM Inspect software, v. 2020 (GOM GmbH, Braunschweig, Germany).

**Figure 4 polymers-13-03469-f004:**
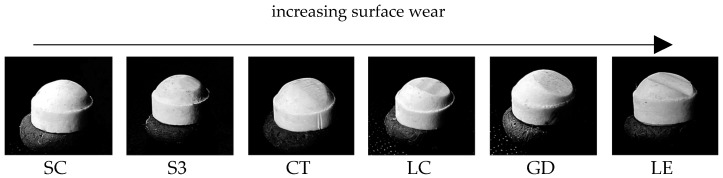
Increasing surface wear of the enstatite antagonists.

**Figure 5 polymers-13-03469-f005:**
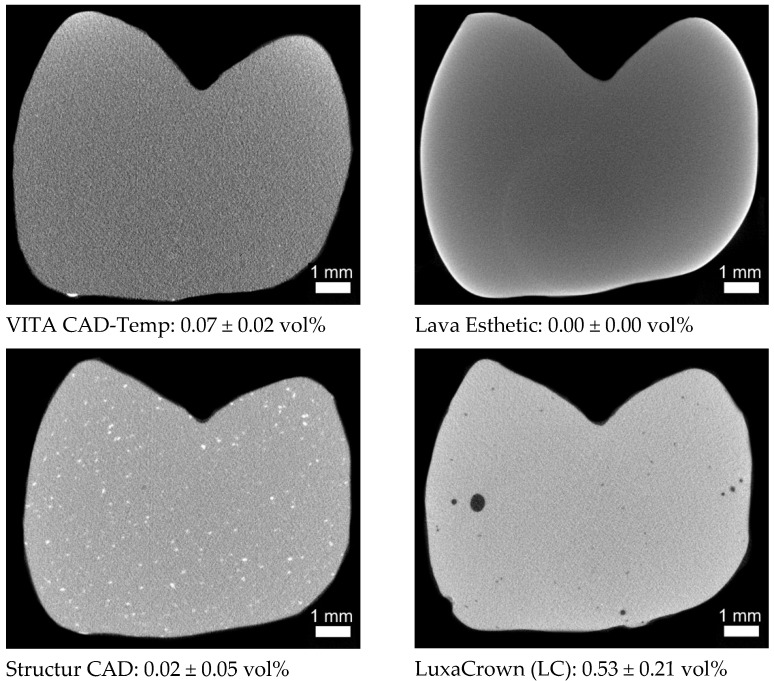
µXCT sectional images from the centre of the three-unit FDPs, each with the total porosity (measured on a region of interest (ROI) 6 mm × 4 mm × 4.2 mm from the centre of the restoration).

**Table 1 polymers-13-03469-t001:** Mechanical properties of tested materials with authorized type of restoration compared to enamel and dentine (* manufacturer’s information).

	Products (Code)	Composition	Micro- Hardness Vickers	Elastic Modulus GPa	Flexural Strength MPa
Bis-acryl composite resins	Structur 3 (S3)	UDMA, Bis-GMA, Filler: Fumed silica (50 nm) [26], 32 wt% [27]	13 [27]	1.9	113 * (3-point) 142 * (biaxial)
	LuxaCrown (LC)	Dimethacrylate Resin 35–45 wt%, filler content 46 wt% (with Ø 0.02–1.5 µm) *	/	/	154 * 110 [28]
Resin composites (CAD/CAM discs)	Structur CAD (SC)	/	/	/	>120 *
	VITA CAD-Temp (CT)	PMMA, 14 wt% microfillers (SiO_2_) [29]	25 [29] 25 [30]	3.62.8 [31]	88.5 [31] (3-point)
	Grandio disc (GD)	Dimethacrylates, 86 wt% glass ceramic filler; functionalized	155 *	18 *	333 *
Zirconia	Lava Esthetic (LE)	5 mol% Yttria-stabilized Cubic Zirconia Polycrystal [32]	1200 *	216 *	800 (3-point) *
Enamel	/	/	313.3 [15]	59.7 [15]	/
Dentin	/	/	62.3 [15]	16.5 [15]	/

**Table 2 polymers-13-03469-t002:** Investigated materials: processing method, indication, product name, code and manufacturer.

Processing Method	Indication	Material	Code	Manufacturer	LOT
**Direct processing material**	Temporary materials	LuxaCrown	LC	DMG GmbH, Hamburg, Germany	791629
		Structur 3	S3	VOCO GmbH, Cuxhaven, Germany	1919450
**Indirect processing material**	Temporary materials	Structur CAD	SC	VOCO GmbH, Cuxhaven, Germany	V77579
		VITA CAD-Temp	CT	VITA Zahnfabrik H. Rauter GmbH & Co. KG, Bad Säckingen, Germany	78210
	Permanent materials	Lava Esthetic	LE	3M Deutschland GmbH, Seefeld, Germany	5364987
		Grandio disc	GD	VOCO GmbH, Cuxhaven, Germany	2006665

**Table 3 polymers-13-03469-t003:** Illustration of the fabrication process of tooth stumps, models, FDPs, and test specimens.

Tooth Stumps	Models	Three-Unit FDP	Test Specimen
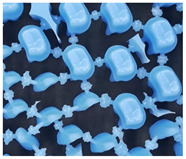	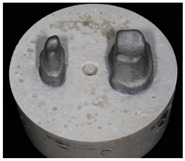	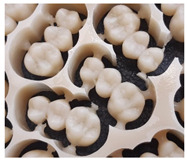	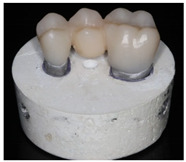
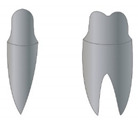	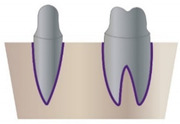	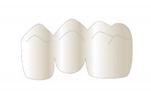	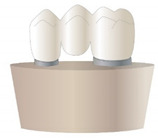

**Table 4 polymers-13-03469-t004:** Fracture load (N): mean values and standard deviations (SD); specimens were grouped according to their indication (directly processed—temporary: S3, LC; indirectly processed – temporary: SC, CT; indirectly processed—permanent: GD, LE) and group (_0 unstressed without CS + TC (control group), _X stressed by CS + TC (experimental group)).

**FDPs**	**Fracture Load**	***p* Value**
S3_0	804.7 (95.3)	0.810
S3_X	850.4 (344.9)	
LC_0	943.3 (217.2)	0.216
LC_X	756.1 (392.4)	
SC_0 ^A^	823.0 (148.4)	0.523
SC_X ^B^	875.0 (190.3)	
CT_0 ^A^	433.5 (67.6)	<0.001
CT_X ^B^	582.8 (57.1)	
GD_0	1099.4 (149.6)	0.021
GD_X ^B^	1326.8 (176.8)	
LE_0	1541.9 (645.8)	0.509
LE_X ^B^	1705.8 (248.1)	

^A^ significant differences in fracture load within an indication without CS + TC. ^B^ significant differences in fracture load within an indication with CS + TC.

**Table 5 polymers-13-03469-t005:** Maximum vertical substance loss (in mm) on the FDP surface (two samples) after chewing simulation in combination with thermocycling.

	Directly Processed		Indirectly Processed/		Indirectly Processed/	
	Temporary		Temporary		Permanent	
FDPs	S3	LC	SC	CT	GD	LE
1	1.52	0.42	0.81	0.87	0.85	0.05
2	- ^1^	0.56	0.55	0.81	0.78	0.07

^1^ The second test specimen failed during ageing.

**Table 6 polymers-13-03469-t006:** Surface roughness values: Ra (in µm) and Rz (in µm) and standard deviations (SD); significant differences of roughness values between the different surface areas (0: unstressed surface prior to chewing simulation; X: stressed surface after chewing simulation; p: pressure point; w: downward movement).

FDPs	Ra (SD)	Rz (SD)	Sign. Diff. (*p* < 0.05)
S3_0	0.665 (0.098)	4.226 (0.645)	A, B, 1, 2
S3_X_p	1.070 (0.258))	6.559 (1.071)	
S3_X_w	1.947 (1.073)	10.752 (4.945)	
LC_0	0.855 (0.080)	5.678 (0.717)	B, C, 2, 3
LC_X_p	0.858 (0.290)	5.696 (1.472)	
LC_X_w	1.472 (0.593)	10.194 (3.843)	
SC_0	0.637 (0.066)	4.254 (0.873)	A, B, C, 2, 3
SC_X_p	0.985 (0.330)	5.585 (0.706)	
SC_X_w	1.695 (0.742)	9.439 (3.407)	
CT_0	0.541 (0.102)	3.558 (0.976)	A, B, C, 1, 2, 3
CT_X_p	2.870 (2.101)	19.138 (6.038)	
CT_X_w	3.531 (0.755)	24.957 (5.037)	
GD_0	0.803 (0.012)	5.769 (0.291)	A, B, 1, 2
GD_X_p	1.398 (0.272)	10.239 (1.318)	
GD_X_w	1.864 (0.857)	11.839 (3.938)	
LE_0	0.791 (0.038)	6.557 (1.853)	A, B, 1, 2, 3
LE_X_p	0.548 (0.033)	3.517 (0.166)	
LE_X_w	0.749 (0.113)	4.924 (1.358)	

^A^ significant differences in Ra between 0 and X_p. ^B^ significant differences in Ra between 0 and X_w. ^C^ significant differences in Ra between X_p and X_w. ^1^ significant differences in Rz between 0 and X_p. ^2^ significant differences in Rz between 0 and X_w. ^3^ significant differences in Rz between X_p and X_w.

## Data Availability

Data is contained within the article.

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
