# Peer review of "CAD/CAM Resin-Based Composites for Use in Long-Term Temporary Fixed Dental Prostheses"

_polymers, 2021, doi:10.3390/polym13203469_

Round 1

Reviewer 1 Report

I have reviewed the manuscript “CAD/CAM resin-based composites for use in long-term temporary fixed dental prostheses” submitted to “Polymers” for publication. In this paper, authors have analysed the performance of CAD/CAM resin-based composites for the fabrication of long-term temporary fixed dental prostheses (FPD) and compared it with commercially available alternative materials for long-term stability. I found this work interesting and fit well with in the scope of this journal. The manuscript needs some major improvements; there are a few suggestions that authors may consider to improve it further:

The use of English language is reasonable, however, there are a number of punctuation and grammatical errors; that should be corrected and rephrased using academic English for a better flow of text for reader.

Abstract: is precisely written, and the aim of the study is mentioned. Please include a conclusive statement in the abstract.

Based in title and the aim, there is confusion; if the materials under research are temporary applications, then why evaluating them for a long-term use?

Introduction; is detailed, compact, covering the background information and the rationale of the study effectively.

Authors simply  presented load and deformation data after testing specimens under universal testing machine. Some important variables can be calculated from these two parameters, load and deflections, elastic modulus.

Section 2.3: should be Mechanical testing?

Similarly, A 3D laser scanner and statistical analysis may have separate subheadings.

In discussion, please include more studies in context; comparing the discussing findings of the present study,?

What are the limitations of this study? And challenges/obstacles in usage of such for the target applications, Should be included.

Author Response

Dear Reviewer 1, 

Thank you very much for your input!

We have tried to implement your comments as well as possible. You find an overview in the table. 

Best regards

Andreas Koenig

Reviewer1

Answer

is precisely written, and the aim of the study is mentioned. Please include a conclusive statement in the abstract.

Based in title and the aim, there is confusion; if the materials under research are temporary applications, then why evaluating them for a long-term use?

The number of words allowed in the abstract section is limited. We included a short conclusive statement in the end of the abstract.

The main topic of the project was the “long-term temporary” (title) and not simple temporary application as provisional restoration or long-term application as definite restoration.

Only at the end of the manuscript, a thesis was included (intended as an outlook) that the best RBC (GD) investigated in the current study “might be used for application in three-unit FDPs. Further laboratory and clinical studies are necessary to corroborate this thesis.” This thesis refers to an application as definite restoration.

“Authors simply presented load and deformation data after testing specimens under universal testing machine. Some important variables can be calculated from these two parameters, load and deflections, elastic modulus.”

That was at the beginning our motivation, but

(1) the deflections depended not only from the material of the FDP, but also from the 1 mm polyether layer (to simulate the resilience of the human periodontium).

(2) the elastic modulus is a mechanic parameter of materials (like strength) and we tested FDP constructions rather than material parameters.

Section 2.3: should be Mechanical testing?

Similarly, A 3D laser scanner and statistical analysis may have separate subheadings.

Yes, this would be an option, but we already have several sections with three sub-titles. We introduced every method with a new paragraph. We hope that the reviewer agrees with this approach.

In discussion, please include more studies in context; comparing the discussing findings of the present study,?

What are the limitations of this study? And challenges/obstacles in usage of such for the target applications, Should be included.

Unfortunately, most references focus on three-unit FDPs fabricated from ceramic materials rather than RBCs as in the present study. The authors included 20 references in the discussion section, which we think is sufficient. The authors hope that the reviewer agrees with this approach.  

A statement referring to the limitations of the current study has been included in the beginning of the conclusion section as recommended.

Reviewer 2 Report

Thanks for the interesting and excellent article, which I think I can accept as is.

Author Response

Dear Reviewer, 

thank you for your time to read the paper!

Kind regards

Andreas

Reviewer 3 Report

General comments

This study investigated the performance of CAD/CAM resin-based composites for long-term temporary three-unit fixed dental prostheses (FPD), comparing directly fabrication composite materials for long-term temporary three-unit FDP, and CAD/CAM resin-based composites and zirconia for permanent three-unit FDP.

The results of this study clarified Structur CAD was suitable for a long-term temporary three-unit FDP on the basis of the results of loading test for failure, two-body wear test, and analysis of the micro-X-ray computer tomograph images of the center of the three-unit FDPs. This information is useful for dental clinicians.

Some questionable and insufficient issues were found in this paper.

Introduction section

Line 58

“blanks (blocs and discs)” should be changed to “blocks or discs”.

Line 73 and 78

“w.-%” should be changed to “wt%”.

Table 1

The authors recommended Structur CAD; however, composition of Structur CAD is not shown in Table 1. Readers may want to know the composition of Structur CAD. The authors should try to show the composition.

I wonder whether long-term temporary FDPs are really necessary for dental treatment or not. I suppose short-term within one-month temporary FDPs are available in regular dental treatment. I could not sufficiently understand the necessity of temporary FDP for long-term (two-years) from the description of the Introduction. The authors had better emphatically describe the clinical situation which is necessary for long-term FDP.         

Materials and Methods section

Line 139

Polyether rubber material was used between the metal root and resin block as an artificial periodontium. The authors set the thickness of the polyether rubber to 1 mm; however, a thickness of human periodontium is approximately 0.2 mm. Please explain the difference setting. Did you confirm the actual thickness of the polyether between the metal root and resin block on the model used in this experiment?

Line 157

“restauration” should be changed to “restoration”.

Line 160-164

The authors explain why the permanent cement was used in this experiment. Although I understand the aim of use of permanent cement to avoid loosing the FDP during chewing simulation, a temporary FDP is always cemented using temporary cement to remove easily after temporary cementation in clinic. How do the authors think the difference between the experimental procedure and clinical situation?      

Figure 1

The number of the experimental group specimens was 60, which is equal to that of control group specimens; however, 12 specimens were used for imaging methods, as a result, different number of specimens between control (60 specimens) and experiment group (48 specimens) was applied for mechanical loading for fracture test. Why did not the authors make the specimens of 72 for the experimental group?   

Line 170

The control means no application of chewing cycles for the specimens, doesn`t it? The authors had better clearly describe the definition of the control.

Line 177

The subtitle of “Methods” may confuse “Experimental procedure” in line 127. Please reconsider this subtitle.

Line 181

“H2O” should be changed to “H2O”.

Line 195

What is “subgroup E2”?

Results section

Line 231

The results of the fracture loading test showed that only S3 failed during chewing cycle test. Their failure modes were unclear in the description. Were they fracture, falling or loosing? The failure rate of S3 was high. Why did not the authors try further experiment for S3 group? Those failure may cause to operational error during fabrication of S3 FDPs.

Table 4

Readers may be not able to understand what “_0” and “_X” mean, because there was no explanation about them.

Line 248-253

There may be same meaning two sentences. Please confirm them.

P-value should be shown by actual value such as P=0.021, when comparing two groups.    

Table 5

Why is the value of number 2 of S3 absent? Was the specimen failed during chewing cycle test?

Figure 5

Abbreviation of each material had better be used. “vol.-%” should be changed to “vol%”.

Discussion section

Line 303

4   . Discussion should be corrected.

Line 333

“CAD-Temp (CT)” had better be changed to “CT”.

Line 417-418

What is “bur debris”? The description of “bur debris” means “smear layer on the FDP surface generated by preparation with CAM”, doesn`t it?

Line 422

“Summery” should be changed to “Summary” or “Conclusion”. “Conclusion” is better.

Line 435

The reference number of [35] should be eliminated.

Line 437

The last sentence of “Further laboratory and clinical studies are necessary to corroborate this thesis.” had better eliminate from conclusion.

Author Response

Dear Reviewer, 

First of all, we would like to thank you for your comments. We have tried to implement your comments. An Overview give the next table. 

Kind regards from Leipzig/Germany

Andreas Koenig

Round 2

Reviewer 1 Report

Many thanks for responding to comments and revising the manuscript. 

Reviewer 3 Report

The manuscript was well revised according to the reviewer`s comments.

There are several issues to be corrected as follows. Please confirm them.

Line 56

"blocs" should be changed to "blocks".

Table 1

32 wt%, 35-45 wt%, 46 wt%, 14 wt% and 5 mol% may be correct description, respectively.

Line 199

Subtitle of  "2-3 Methods" had better be eliminated.

Line 33

"4. Discussion" is correct.

Line 451

"5. Conclusion" is correct.